# New Neutrophil Parameters in Diseases with Various Inflammatory Processes

**DOI:** 10.3390/biomedicines12092016

**Published:** 2024-09-04

**Authors:** Elżbieta Rutkowska, Iwona Kwiecień, Agata Raniszewska, Rafał Sokołowski, Joanna Bednarek, Karina Jahnz-Różyk, Andrzej Chciałowski, Piotr Rzepecki

**Affiliations:** 1Laboratory of Hematology and Flow Cytometry, Department of Internal Medicine and Hematology, Military Institute of Medicine-National Research Institute, 04-141 Warsaw, Poland; ikwiecien@wim.mil.pl (I.K.); araniszewska@wim.mil.pl (A.R.); 2Department of Internal Medicine, Pneumonology, Allergology, Clinical Immunology and Rare Diseases, 04-141 Warsaw, Poland; rsokolowski@wim.mil.pl (R.S.); jbednarek@wim.mil.pl (J.B.); kjrozyk@wim.mil.pl (K.J.-R.); 3Department of Internal Medicine, Infectious Diseases and Allergology, 04-141 Warsaw, Poland; achcialowski@wim.mil.pl; 4Department of Internal Medicine and Hematology, Military Institute of Medicine-National Research Institute, 04-141 Warsaw, Poland; przepecki@wim.mil.pl

**Keywords:** hematology analyzer, neutrophil scattering items, Sysmex, neutrophil parameters, immature granulocytes, lung cancer, sarcoidosis, COVID-19

## Abstract

The neutrophils evaluation seems interesting in the initial qualifications of patients with various inflammatory processes. In this study, we presented analysis of neutrophils and new parameters of the complexity (NEUT-GI, NE-WX), maturation (IG), size (NE-FSC, NE-WZ), and neutrophil activities (NEUT-RI, NE-WY) in coronavirus disease 2019 (COVID-19), lung cancer (LC), sarcoidosis (SA), and healthy controls (HCs). Peripheral blood (PB) was collected. The new parameters were examined by the Sysmex XN-1500. The mean absolute value for the IG parameter was the highest in the LC group. The differences in NEUT-RI value between COVID-19 and the HC group were observed. No significant differences were noticed between groups in the NEUT-GI granularity parameter. Neutrophil size assessed by NE-FSC parameter was reduced in all groups compared to HCs. The values of complexity (NE-WX), fluorescence (NE-WY), and size (NE-WZ) were the lowest in the HCs, whereas the highest median proportions of NE-WX, NE-WY, and NE-WZ were in LC patients. Patients from the SA group differed significantly from the HC group only for the NE-WZ parameter. We showed the usefulness of neutrophil parameters and their reactivity, morphology, and exhaustion. A more detailed analysis of blood counts may reveal trends that indicate a disease-specific immune response.

## 1. Introduction

In recent times, the view on the role of neutrophils in diseases has changed significantly. Neutrophils were thought to act as an indicator of the early innate immune response by phagocytosing foreign particles. It is now known that activated neutrophils, by secreting various pro-inflammatory cytokines and surface and adhesion molecules, act even as presenting cells and activate T lymphocytes [1]. Neutrophils are the most numerous forms of white blood cells (WBCs). They are of great importance in the innate immune response by phagocytosing bacteria and other pathogens. The disease microenvironment may have a major impact on the morphological and functional diversity of this population. Circulating neutrophils vary based on parameters like cell-surface markers, buoyancy, maturity, and localization. This diversity occurs in both normal conditions and in abnormality, including cancer, infections, and inflammatory disorders [2]. Comparing neutrophils across different disease entities is fascinating and clinically relevant. By studying their behavior in various contexts, we learn more about the mechanisms of the disease, find new therapeutic targets, and indicate other diagnostic determinants.

Neutrophils, as part of the innate immune response, participate in the elimination of the effects of induced infections and maintaining tissue homeostasis. Neutrophil counts and activation status can serve as diagnostic and prognostic markers for various diseases. For example, elevated neutrophil levels may indicate infection, inflammation, or tissue damage [3,4,5].

Monitoring changes in neutrophil populations over time can help track disease progression and response to treatment. Dysregulated neutrophil responses contribute to immunopathology in conditions such as sepsis, autoimmune diseases, and chronic inflammatory disorders [6,7]. Moreover, understanding the mechanisms behind neutrophil activation and recruitment can guide therapeutic strategies.

Interestingly, neutrophils may play a variety of roles in cancer. They interact with cancer cells, promoting cancer initiation, growth, and metastasis, contributing to cancer progression. On the other hand, antitumor neutrophils are essential in immune surveillance against cancer [8,9,10]. They release toxic substances within the tumor microenvironment, aiding immune defense, but they can also succumb to tumor influence, promoting immune evasion. For example, neutrophil granules (like elastase and vascular endothelial growth factor) impact tumor cell proliferation, metastasis, and angiogenesis [11]. A recent discovery revealed that neutrophils actively fight solid tumors. When T cells (immune cells) attack tumors cells, they activate neutrophils to eliminate cancer cells together [12]. Their role goes beyond inflammation, and understanding this could improve cancer treatments.

It is well known that neutrophils are an important line of innate defense, and their role is extremely important, although knowledge about their force in the course of sarcoidosis is not yet fully understood. The cause of sarcoidosis is unknown; it may occur in people with a genetic predisposition and trigger an immune response against an antigen, leading to the formation of granulomas [13]. Under the influence of interleukin 8, secreted by monocytes and macrophages, neutrophils come to the site of inflammatory nodule formation [14].

Coronavirus disease 2019 (COVID-19) arises from severe acute respiratory syndrome coronavirus 2 (SARS-CoV-2) infection. The disease can progress asymptomatically, ranging from minor symptoms to respiratory and multi-organ failure, leading to death. Dysregulation of neutrophils is associated with cytokine storms, tissue injury, and thrombotic events. Circulating neutrophils can form low-density structures (LDNs) as well as neutrophil extracellular traps (NETs). Their increased formation correlates with disease severity and poor prognosis in COVID-19 patients. Severe COVID-19 shows an increase in immature neutrophil populations with recent activation features. Promising attempts aim to better assess the role of neutrophils and modulate their reactivity in COVID-19 patients [15].

The purpose of this study was to analyze neutrophils and assess new parameters related to the complexity, size, and activity of neutrophils in various disease states involving inflammation. We have selected three diseases that differ in the course of the inflammatory process: sarcoidosis as a disease with chronic inflammation, SARS-CoV-2 virus infection causing a severe immune reaction, and inflammation in the course of lung cancer. Selected patients were compared to a healthy control group (HC). The number of neutrophils, the state of activation, and potential exhaustion were assessed to determine their role in the course of the diseases, as well as to highlight the utility of new hematological parameters in the initial evaluation of patients.

## 2. Materials and Methods

### 2.1. Patients

The research group included patients with various disease states in which the inflammatory process is an important element. There were 33 patients with active COVID-19 infection, 33 patients with lung cancer, 34 patients with sarcoidosis, and 28 HCs. Samples were collected in three clinics of the Military Medical Institute-National Research Institute from May 2020 to December 2023 at the Department of Internal Medicine and Hematology and the Department of Infectious Diseases and Allergology and Clinic of Internal Diseases, Pneumonology, Allergology, Clinical Immunology and Rare Diseases. Blood samples were then delivered to the Laboratory of Hematology and Flow Cytometry for analysis. The patients are characterized in Table 1.

The routine blood count values and proportions, such as absolute counts of WBC, neutrophils, lymphocytes, monocytes, eosinophils, and basophils, were compared between three diseases and HCs.

### 2.2. Materials

Peripheral blood (PB) samples from all patients were taken. The samples were obtained using EDTA-K3 tubes (Beckton Dickinson, Franklin Lakes, NJ, USA). Processing was performed immediately upon receipt of the sample using a Sysmex XN-series hematology system (Sysmex Co., Kobe, Japan). All parameters tested were assessed from whole blood. The blood samples from patients participating in this study were collected during routine examinations. All patients gave informed consent (Medical Chamber in Warsaw: KB/1441/23/: cancer patients; Military Institute of Medicine Ethics Committee number: 47/ WIM/2020: COVID-19 patients, Military Institute of Medicine Ethics Committee: 25/WIM/2018: sarcoidosis patients).

### 2.3. Methods: New Neutrophil-Related Sysmex Parameters

Cellular analysis using the SYSMEX analyzer is based on the principle of fluorescence cytometry using a semiconductor laser and measuring forward and side scatter, as well as side scatter of fluorescent light. Measurement of parameters characterizing neutrophils is also based on the principles of fluorescence flow cytometry. Cell size, structural differentiation, and fluorescence intensity are assessed. Upon activation, cells change the composition of lipid particles on their plasma membrane and exhibit different cytoplasmic differentiation than resting cells, resulting in higher fluorescence intensity.

The two available research parameters assessing neutrophil granularity intensity (NEUT-GI) and neutrophil reactivity index (NEUT-RI) indicate ongoing inflammation [16,17]. NEUT-GI assesses the light scattering intensity, indicating the state of cytoplasmic granulation, altered by increased granularity or vacuolization. The NEUT-RI parameter assesses the fluorescence intensity (FI), which may indirectly reflect the metabolic activity of the neutrophil population [17], and represents the RNA content in neutrophils. To directly evaluate the metabolic activity of neutrophils, it would be necessary to perform functional tests on neutrophils. NEUT-RI as an early marker of innate immune response may correlate with sepsis [18]. During cell activation, pro-inflammatory signals are initiated leading to changes in the composition of nucleic acids and cell organelles. The fluorescent agent combines with these structures and, under the influence of laser light, emits a higher intensity of light. Increased metabolic activity of cells is manifested by an increase in the NEUT-RI parameter [19].

The population of immature granulocytes including promyelocytes, myelocytes, and metamyelocytes is described by the IG parameter [20].

The NE-WX, NE-WY, and NE-WZ parameters are determined based on the spread around the average fluorescence intensity. They show the range of fluorescence values for cells without cases below 20% of the highest level for the distribution curve. The values correspond to the size, width, and complexity of neutrophils, respectively [16]. A brief description of the parameters is presented in Table 2 and Figure 1.

### 2.4. Statistical Analysis

Statistical analyses were performed using Statistica 13.0 software (TIBCO Software, Palo Alto, CA, USA). *p* values below 0.05 were considered statistically significant. Results are expressed as medians (Q1–Q3). The Kruskal–Wallis ANOVA test and post hoc analysis test were used to compare groups.

## 3. Results

The characteristics of the studied population are described in Table 1. The average age of all examined patients and HCs was 55 years. Table 1 also includes the stage of advancement for SA and LC patients and contains important parameters for patients with COVID-19 and SA. For LC, we also presented histological cancer subtypes.

### 3.1. Basic Blood Count Tests

In order to assess the basic distribution of leukocytes, the examined patients underwent a basic blood count. WBC, neutrophil, lymphocyte, monocyte, eosinophil, and basophil counts and IG values were analyzed. The significantly highest WBC value was obtained in the LC group (8.31 × 10^3^/µL). The remaining groups did not differ from each other. A similar relationship was obtained for the number and percentage of neutrophils, with the highest value for the LC group than COVID-19, SA, and HC (5.62 vs. 2.89 vs. 3.61 vs. 3.63; *p* < 0.001, respectively). The lowest absolute count of lymphocytes was observed in the COVID-19 group significantly in relation to the HC group (1.23 vs. 1.63, *p* = 0.0137, respectively) but without statistical significance compared to the SA and HC groups. We also recorded the lowest median proportion of monocytes, eosinophils, and basophils for this group of patients, with no differences between the other groups. The mean absolute value for the IG parameter was significantly the highest in the LC group (0.04 vs. 0.02 for other groups, *p* = 00014, respectively) but with no significant difference in comparison to the COVID-19 group (Table 3 and Figure 2).

### 3.2. New Neutrophil-Related Sysmex Parameters

The values of hematological parameters were determined for the studied patients (description of new hematological neutrophil parameters in Table 2, Materials and Methods section). When we analyzed the new neutrophil parameters, we observed differences in the NEUT-RI value between COVID-19 and the HC group (46.1 vs. 48.3, *p* = 0.0273). No significant differences were observed between groups in the assessment of the NEUT-GI granularity parameter, while the NEUT-GI to NEUT-RI ratio turned out to be the lowest for the HC group (3.17) compared to the COVID-19 and SA groups (3.31, 3.29, *p* > 0.0229). Cell size assessed by the median of the NE-FSC parameter was reduced in all tested groups (COVID-19 vs. LC vs. SA, respectively) compared to HCs (87.9 vs. 90.5 vs. 88.3 vs. 94.6 in HC group, *p* < 0.001).

The values corresponding to the distribution of complexity (NE-WX), fluorescence (NE-WY), and size (NE-WZ) of neutrophils were the lowest in the HCs. We noticed the highest median proportion of NE-WX, NE-WY, and NE-WZ parameters in LC patients with a significant difference to the HC group (317.0 vs. 302.5, *p* = 0.0323; 642.0 vs. 584.5, *p* = 0.0001; 664.0 vs. 541.0, *p* < 0.0001). Patients from the SA group differed significantly from the HC group only for the NE-WZ parameter (646.0 vs. 541.0, *p* < 0.0001) (see Table 4 and Figure 3).

## 4. Discussion

The new technological possibilities of the hematology analyzer allow quick and low-cost assessment of the degree of activation, stability, and possible depletion of cells that affect the patient’s immune status. Parameters determined during routine morphology provide new information to estimate the level of cell activity and maturity [16,21,22].

Basic morphology tests have already revealed differences between the selected groups. We found a significant increase in WBC count, number of lymphocytes, neutrophils, and monocytes, as well as immature forms of granulocytes in patients with LC. We confirmed characteristic lymphopenia and reduced the eosinophil and basophil counts in patients with COVID-19, whereas the results of SA patients did not differ from those of HCs. The basic morphological results obtained in the studied groups did not turn out to be surprising. It is known that patients with COVID-19 manifest lymphopenia, neutrophilia, and reduced eosinophil values [23,24]. In our case, patients were classified in the medium severe group; therefore, we did not find significant neutrophilia. Markers of SA in PB are still understudied. The most common location of sarcoidosis concerns the lungs and lymph nodes inside the chest, and noticeable changes in the cellular composition are visible in the bronchoalveolar lavage fluid (BALF) [25]. A direct dependence between routine blood tests and cancer diagnosis has not yet been found. Although, the link between inflammation and cancer development is an established concept, and chronic inflammation consisting of excessive numbers of T lymphocytes, NK cells, and neutrophils may persist. Therefore, chronic inflammation may be pathogenic in LC [26].

In the next stage of research, we examined neutrophils by using new hematological parameters determining their cellular status: IG, NEUT-RI, NEUT-GI, NE-FSC, and NE-WX, -WY, -WZ.

The IG parameter indicating immature forms of cells of the granulocytic line seems to be an interesting one. We found an increased IG level in patients with LC compared to the SA and HC groups. The remaining groups did not differ from each other. The increased number of IGs may be related to a deficiency of mature, well-functioning neutrophils. It has been shown that cancer cells are able, through various mechanisms, to enhance the proliferation of bone marrow cells and the appearance of immature forms in the periphery [27,28]. Identification of immature granulocytes may be an indicator of cancer-related bone marrow proliferation. It allows for early detection of bone marrow hyperplasia associated with cancer [29]. In our previous study in patients with myelodysplastic syndromes, we showed that the median number of IGs was higher in patients with detected mutations than without cytogenetic changes [30]. Lu et al. showed that IG assessment is important in the diagnosis of myeloid malignancies and could be helpful in screening [31]. Viral infections, sepsis, or active disease states may activate circulating mature neutrophils and further induce granulopoiesis [32]. This mechanism causes the presence of an increased number of immature neutrophils in the periphery, which may have immunosuppressive or pro-inflammatory effects [33,34]. An increased number of neutrophils can also be observed in patients with active COVID-19 disease [35].

A recent research trend is the evaluation of ways to target neutrophils and NET formation for potential therapeutic interventions [36]. Our observations and attempt to use new neutrophil-related parameters as a measurable way to assess neutrophils do not assess the formation of NETs, but they fit well into this trend.

Interestingly, the role of neutrophils in sarcoidosis is unknown. In our study, we did not find any differences in the number and percentage of granulocytes or the amount of IGs in the SA group compared to the control group. Discovering the role of neutrophils in this disease requires a more detailed examination of the activation and stimulation state of these cells. An increased percentage of neutrophils in BALF in the case of newly diagnosed pulmonary SA was associated with the need for steroid treatment and could be a marker indicating disease progression [37]. BALF neutrophil count was significantly higher in patients with advanced disease and radiological grade II or III SA [38,39]. An increase in the level of neutrophil to lymphocyte ratio (NLR) in SA patients was noticed by Almadari M.G. et al., and this parameter was recommended as a guide for diagnosis, detection of disease severity, and involvement of the lung parenchyma [40].

Hematological analyzers also provide information on the size of neutrophils (NEUT-FSC, neutrophil reactivity; NEUT-RI, the complexity of the cytoplasm structure; NEUT-GI, the integrity of the population of these cells; NE-WX, -WY, -WZ), which, as we prove here, are worth using in the screening of patients with various diseases.

It is interesting that the activation value measured by the NEUT-RI parameter is reduced in all tested groups compared to the HCs. In our previous studies on COVID-19, we found that convalescent patients showed significantly lower NEUT-RI values compared to patients with active infection [35], and similarly to the current study, this parameter had the highest value in healthy people. Perhaps a low NEUT-RI value may indicate the exhaustion of neutrophils during inflammation and after the disease. In healthy people, an increased NEUT-RI value may indicate the proper functionality of neutrophils and their ability to defend against inflammation [41]. The inflammatory biomarker NEUT-RI assessed in the early diagnosis of sepsis may play an important role in predicting outcome, supporting clinical decisions [42,43]. Others have shown that the NEUT-RI parameter can provide additional information regarding the indication of sepsis and the prediction of mortality in a pediatric ward [18]. Assessment of the NEUT-RI value together with examination of clinical symptoms and other inflammatory markers may help in the initial diagnosis. Cell granularity measured with the NEUT-GI parameter turned out to be insignificant in selected research groups. We found no differences between the groups and in relation to the control group. This parameter seems to be irrelevant or too insensitive to capture changes in the cell. Similarly, we did not find differences in the NEUT-GI and NEUT-RI parameters in the group of MDS patients divided according to the presence of mutations [30].

However, the combination of both above-mentioned parameters in the NEUT-GI/NEUT-RI ratio may help in identifying patients with an acute course of the disease: COVID-19 and LC. There was a significant increase in both groups in comparison to the healthy one. Therefore, it is recommended to demonstrate this relationship in screening tests.

Additionally, we analyzed the NE-FSC parameter reflecting cell size. The NE-FCS value was significantly the highest in HCs, with no significant differences between other groups. In the case of sepsis, viral infections, or bacterial infections, smaller sizes of granulocytes assessed by the NE-FSC parameter were also observed [44,45,46]. The NE-FSC parameter was used to evaluate cases of myelodysplastic syndrome with reduced neutrophilic granules. The authors found that the evaluation of this index is useful in detecting cases with reduced neutrophil granule counts [47]. The reduction in neutrophil size may indicate exhaustion after degranulation, which may explain this phenomenon. Neutrophil activation leads to structural changes enabling phagocytosis and the synthesis of pro-inflammatory cytokines and thus changes cell size [48].

We showed significant differences between groups in terms of neutrophil population complexity. They are not homogeneous in size, functionality, and complexity of structure in all study groups. As expected, the most uniform neutrophil population was observed in the control group and the most differential in LC patients. LC patients showed a higher percentage of NE-WX and NE-WY than in patients with SA and the control group. NE-WX assesses the intrinsic complexity of neutrophils based on their granularity, and elevated values may indicate specific morphological changes. NE-WY assesses the RNA/DNA content of neutrophils; therefore, changes in NE-WY may provide insight into cellular activity. Although there are no specific studies directly evaluating NE-WX and NE-WY in patients with LC and SA, there are valuable studies examining neutrophil parameters in various contexts [19,49]. The NE-WX indicator has been studied in MDS, providing additional information to differentiate macrocytic anemia from other conditions [50]. In our previous COVID-19 study, we examined neutrophil parameters to distinguish convalescent patients from those with active SARS-CoV-2 infection [35]. These parameters have been found to be useful tools for assessing neutrophil activity during infection and recovery.

We investigated the use of newer features of automated hematology analyzers for the assessment of neutrophil populations and obtained promising results in patient stratification. We were looking for quick and easily available markers enabling the initial diagnostic assessment of the patient. The role of the examined parameters is clearly visible in the group of cancer patients and those with viral infections. In the case of patients with SA, their application does not seem to be important.

The limitation of this study is the number of patients and the lack of division into disease stage groups, and although the method used is cheap and fast, it cannot be a key laboratory technique for detecting neutrophil abnormalities. A limitation of the work may be the fact that the results may be slightly distorted by the process of NETosis in the course of cancer, compared to the healthy state. NETosis affects the binding of the dye to DNA and also changes the shape of neutrophils [51]. Additionally, in viral infection, neutrophils often form neutrophil–platelet aggregates, and their presence most likely leads to changes in the width/height of the Sysmex parameters, which is a coefficient for the Sysmex data set [52]. Finally, the underlying drug therapy in these patients may lead to impaired neutrophil function, also introducing additional bias into the results. However, further studies on larger samples, stratified by stage of progression, are necessary to confirm our suggestions and enable the clinical usefulness of hematological indicators in prognosis of the course of the disease.

Future work should be undertaken to design models that analyze trends observed in blood counts using the study parameters discussed in this study.

## 5. Conclusions

In the current study, we showed the usefulness of parameters assessing the neutrophil population, their reactivity, morphology, and exhaustion. We hypothesize that research parameters assessed as part of a routine blood count may identify disease-specific inflammation. In a small blood sample, we can indicate markers of diseases such as viral infections, cancer, or auto-inflammatory processes. Moreover, as technology advances, testing blood cell parameters is becoming more efficient, allowing for faster and more insightful results. The ease, speed, and availability of basic tests make them an essential tool in the diagnosis and treatment of various diseases.

## Figures and Tables

**Figure 1 biomedicines-12-02016-f001:**
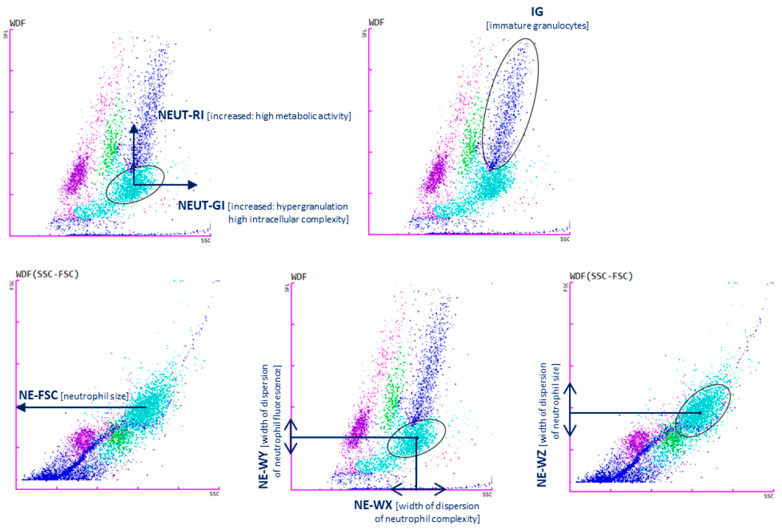
Cell analysis with Sysmex XN-1500: neutrophil reactivity intensity parameter (NEUT-RI determined by an increased shift from the lateral fluorescence signal SFL: metabolic activities with RNA content, on the *y*-axis; neutrophil granularity intensity (NEUT-GI) determined by increased shift from side scatter SSC: intracellular structure on the *x*-axis; immature granulocytes (IGs); NE-FSC neutrophil forwards scatter, indicated by FSC vs. SSC corresponds to the size of neutrophil cells; NE-WX corresponds to the complexity of the cell population, with respect to SSC; NE-WY represents the fluorescence distribution width of cells population; NE-WZ reflects the distribution width of cells population, proportional to the width of dispersion of cell size. Plots show the distribution of leukocytes: lymphocytes (purple), monocytes (green), mature granulocytes (light blue), immature granulocytes (navy blue). Each dot represents one cell.

**Figure 2 biomedicines-12-02016-f002:**
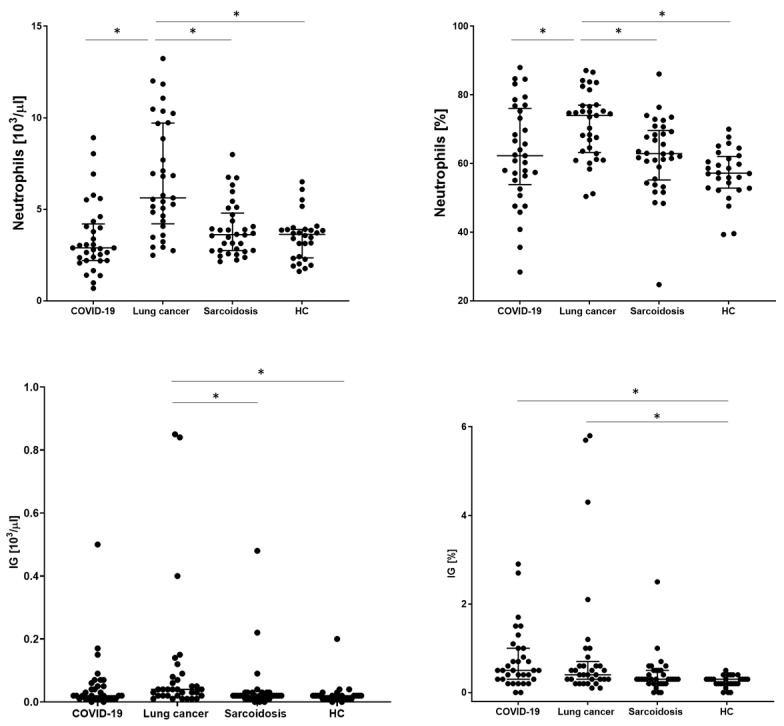
The differences in the proportion of count of neutrophils and immature neutrophils (IG) between patients with COVID-19, lung cancers, sarcoidosis, and healthy controls (HCs). Data expressed as median with interquartile range. * Indicates *p* is statistically significant.

**Figure 3 biomedicines-12-02016-f003:**
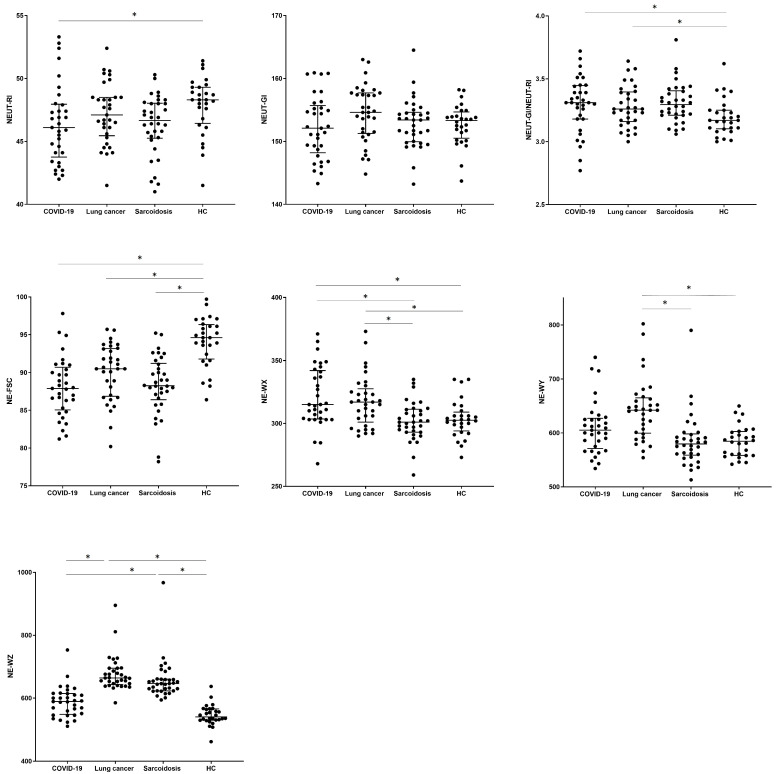
The differences in the percentage of parameters measured with the Sysmex device for neutrophils between patients with COVID-19, lung cancer, sarcoidosis, and a control group of healthy people (HC). Data expressed as median with interquartile range. * Indicates *p* is statistically significant. Abbreviations: NE-FSC, neutrophil size; NE-SFL, neutrophil fluorescence intensity (NEUT-RI, neutrophil reactivity intensity); NE- SSC, neutrophil complexity (NEUT-GI, neutrophil granularity intensity); NE-WX, width of dispersion of neutrophil complexity; NE-WY, width of dispersion of neutrophil fluorescence; NE-WZ, width of dispersion of neutrophil size.

**Table 1 biomedicines-12-02016-t001:** Patient characteristics.

	SA	COVID	LC	HC
Number of patients	34	33	33	28
Sex F/M (n)	9/25	12/21	19/14	25/3
Age (mean ± SD years)	45.2 ± 13.0	58.0 ± 17.5	67.0 ± 8.5	50 ± 11.3
Stage I/II/III/IV	11/23/ n/a/ n/a	n/a	2/4/17/10	n/a
LC histological subtype	n/a	n/a	SCLC 20, 60.6% SQCLC 4, 30.8% ADC 7, 30.8% NOS 1, 7.7% LCC 1, 7.7%	n/a
DLCO (>80%/<80%) (n/n)	17/17	n/a	n/a	n/a
Saturation (mean ± SD%)	n/a	91.0 ± 7.5%	n/a	n/a
Conventional (passive) oxygen therapy (n,%)	n/a	7, 30.4%	n/a	n/a
Mechanical ventilation therapy (n,%)	n/a	3, 13.0%	n/a	n/a

Abbreviations: ADC, adenocarcinoma; COVID-19, coronavirus disease 2019; DLCO, diffusing capacity of the lungs for carbon monoxide; F, female; M, male: HC, healthy control; LC, lung cancer; LCC, large cell carcinoma; NOS, not otherwise specified; SA, sarcoidosis; n/a—not applicable; SCLC, small cell lung cancer; SQCLC, squamous cell lung cancer.

**Table 2 biomedicines-12-02016-t002:** The parameters determined by hematological analyzer in peripheral blood (PB) samples.

Parameter	Parameter Description
NEUT-RI/NE-SFL	The mean value of fluorescence intensity; reflects metabolic activity of neutrophils
NEUT-GI/NE-SSC	Provides information about the density or complexity of the cell and depicts the granularity of the cells
IG	Immature granulocytes
NE-FSC	Intensity of frontally scattered light; neutrophil size
NE-WX	Laterally scattered light intensity; width of dispersion of neutrophil complexity
NE-WY	Intensity of fluorescent light; width of dispersion of neutrophil fluorescence
NE-WZ	Intensity of frontally scattered light; width of dispersion of neutrophil size

**Table 3 biomedicines-12-02016-t003:** Proportion of Sysmex morphological parameters in patients with COVID-19 (A), lung cancer (B), sarcoidosis (C), and control group (HC) (D). Data expressed as median (Q1–Q3). * Indicates *p* is statistically significant.

Hematological Parameters	COVID-19 (A) Median (Q1–Q3)	Lung Cancer (B) Median (Q1–Q3)	Sarcoidosis (C) Median (Q1–Q3)	Healthy Control (D) Median (Q1–Q3)	* *p* < 0.05 Group A-B-C ANOVA, Kruskal–Wallis	* *p* < 0.05 Group, in Groups Post Hoc
WBC [10^3^/µL]	4.58 (3.89–6.81)	8.31 (6.50–11.87)	5.73 (4.89–7.46)	5.95 (4.78–6.53)	* *p* < 0.001	A-B * *p* < 0.0001 B-D * *p* = 0.0053 B-C * *p* = 0.0050
NEUTROPHILS [10^3^/µL]	2.89 (2.20–4.06)	5.62 (4.35–9.69)	3.61 (2.75–4.70)	3.63 (2.36–3.90)	* *p* < 0.001	A-B * *p* < 0.0001 B-C * *p* = 0.0031 B-D * *p* = 0.0002
LYMPHOCYTES [10^3^/µL]	1.23 (0.70–1.57)	1.48 (1.17–1.99)	1.39 (1.03–1.59)	1.63 (1.30–2.08)	* *p* = 0.0137	A-D * *p* = 0.0016
MONOCYTES [10^3^/µL]	0.40 (0.27–0.64)	0.62 (0.49–0.78)	0.53 (0.42–0.68)	0.54 (0.36–0.65)	* *p* = 0.0100	A-B * *p* = 0.0073
EOSINOPHILS [10^3^/µL]	0.06 (0.00–0.10)	0.08 (0.04–0.14)	0.15 (0.03–0.06)	0.12 (0.08–0.19)	* *p* < 0.001	A-C * *p* < 0.0001 A-D * *p* = 0.0127
BASOPHILS [10^3^/µL]	0.02 (0.01–0.02)	0.04 (0.02–0.05)	0.04 (0.03–0.06)	0.04 (0.02–0.04)	* *p* <0.001	A-B * *p* = 0.0002 A-C * *p* < 0.0001 A-D * *p* = 0.0026
IG [10^3^/µL]	0.02 (0.01–0.05)	0.04 (0.02–0.07)	0.02 (0.01–0.03)	0.02 (0.01–0.02)	* *p* = 0.0014	B-C * *p* = 0.0250 B-D * *p* = 0.0016
PLT [10^3^/µL]	210 (177–292)	248 (184–308)	238 (186–308)	249 (213–278)	*p* = 0.7209	-
NEUTROPHILS [%]	62.3 (55.1–75-3)	74.0 (63.6–76.9)	62.9 (55.5–69.3)	57.2 (52.8–61.8)	* *p* < 0.001	A-B * *p* = 0.0259 B-C * *p* = 0.0224 B-D * *p* < 0.0001
LYMPHOCYTES [%]	26.9 (18.1–34.0)	18.6 (14.8–25.0)	22.1 (18.8–30.6)	29.7 (25.8–33.4)	* *p* = 0.001	A-B * *p* = 0.0171 B-D * *p* < 0.0001
MONOCYTES [%]	8.2 (5.5–10.3)	7.7 (6.2–9.1)	8.5 (7.2–11.5)	8.6 (6.7–9.9)	*p* = 0.2311	*-*
EOSINOPHILS [%]	0.9 (0.0–2.2)	0.9 (0.3–2.4)	2.1 (1.6–4.1)	2.1 (1.2–3.1)	* *p* < 0.001	A-C * *p* < 0.0001 A-D * *p* = 0.0164 B-C * *p* = 0.0007
BASOPHILS [%]	0.4 (0.2–0.5)	0.6 (0.3–0.8)	0.6 (0.4–1.0)	0.6 (0.5–0.8)	* *p* = 0.001	A-C * *p* < 0.0001 A-D * *p* = 0.0049
IG [%]	0.5 (0.3–1.0)	0.4 (0.3–0.6)	0.3 (0.2–0.5)	0.3 (0.2–0.3)	* *p* = 0.002	A-D * *p* = 0.0004 B-D * *p* = 0.0038

Abbreviations: HC, healthy control; IGs, immature granulocytes; PLTs, platelets; WBC, white blood cell count.

**Table 4 biomedicines-12-02016-t004:** Differences in the proportion of new hematological parameters connected with neutrophils in peripheral blood between patients with COVID-19 (A), lung cancers (B), sarcoidosis (C), and healthy patients (D). Data expressed as median (Q1–Q3). * Indicates *p* is statistically significant.

Hematological Parameters	COVID-19 (A) Median (Q1–Q3)	Lung Cancer (B) Median (Q1–Q3)	Sarcoidosis (C) Median (Q1–Q3)	Healthy Control (D) Median (Q1–Q3)	* *p* < 0.05 Group A-B-C ANOVA, Kruskal–Wallis	* *p* < 0.05 Group, in Groups Post Hoc
NEUT-RI [FI] or NE-SFL [ch]	46.1 (44.1–47.9)	47.1 (45.6–48.5)	46.6 (45.3–48.0)	48.3 (46.6–49.3)	* *p* = 0.0273	A-D * *p* = 0.0428
NEUT-GI [SI] or NE-SSC [ch]	152.1 (146.7–155.3)	154.6 (151.4–157.7)	153.4 (150.0–154.5)	153.3 (150.8–154.4)	*p* = 0.1649	-
Ratio NEUT-GI/NEUT-RI	3.31 (3.18–3.46)	3.26 (3.16–3.39)	3.29 (3.21–3.40)	3.17 (3.10–3.25))	* *p* < 0.0229	A-D * *p* = 0.0396 C-D * *p* = 0.0404
NE-FSC [ch]	87.9 (85.3–90.4)	90.5 (87.0–93.1)	88.3 (86.6–91.1)	94.6 (92.0–96.2)	* *p* < 0.001	A-D * *p* < 0.0001 B-D * *p* = 0.0027 C-D * *p* < 0.0001
NE-WX	315.0 (304.0–341.0)	317.0 (304.0–325.0)	301.0 (293.0–311.0)	302.5 (295.0–308.0)	* *p* = 0.001	A-C * *p* = 0.0015 A-D * *p* = 0.0072 B-D * *p* = 0.0323 B-C * *p* = 0.0086
NE-WY	605.0 (575.0–625.0)	642.0 (601.0–663.0)	579.5 (560.0–597.0)	584.5 (559.0–602.0)	* *p*< 0.001	B-C * *p* < 0.0001 B-D * *p* = 0.0001
NE-WZ	589.0 (551.0–614.0)	664.0 (646.0–695.0)	646.0 (623.0–660.0)	541.0 (530.5–565.0)	* *p*< 0.001	A-B * *p* < 0.0001 A-C * *p* = 0.0001 B-D * *p* < 0.0001 C-D * *p* < 0.0001

Abbreviations: NE-FSC, neutrophil size; NE-SFL, neutrophil fluorescence intensity (NEUT-RI, neutrophil reactivity intensity); NE- SSC, neutrophil complexity (NEUT-GI, neutrophil granularity intensity); NE-WX, width of dispersion of neutrophil complexity; NE-WY, width of dispersion of neutrophil fluorescence; NE-WZ, width of dispersion of neutrophil size.

## Data Availability

The data presented in this study are available in this article, further inquiries can be directed to the corresponding author.

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
