# Peer review of "New Neutrophil Parameters in Diseases with Various Inflammatory Processes"

_biomedicines, 2024, doi:10.3390/biomedicines12092016_

Round 1

Reviewer 1 Report

Comments and Suggestions for Authors

The presented paper focuses on the possibility of Sysmex-based analyzing neutrophils under different type of diseases. Since neutrophils are a key player in innate immune response I went over the draft with very high interest. I spotted some problems/limitations that must be addressed during the peer-review process, namely: 

1) It is not clear how the blood was processed before the analysis. I understand that the blood was drawn from three different units (based on the disease/ethical committee approval) and then analyzed in a single unit. Please elaborate on this workflow since it is crucial for the results reliability. 

2) I don't find the exact methodology for neutrophils staining. Since Authors refer to fluorescence parameter NE-WY, it is not clear how the fluorescence happens - whether there is a neutrophil-specific staining (eg Ly6G) or any subcellular/membrane staining? It needs to be clarified.

3) Also, I don't agree that provided data on sysmex-based is novel. This system has been introduced to the market in 2000's and have already 3500+ published papers (and 276 focusing on WBC abnormalities, 112 on neutrophils under different codnitions). 

4) Also, it is not clear whether the data represents the results from the whole blood or isolated/purified neutrophils.

5) There is no information given on the statistical approach. Table 1 for sure shows the difference in terms of sex and age parameters. This introduces additional bias to the paper.

6) Why did the Authors choose to show the data as a Q1-Q3 intervals? It is quite a unique approach.

7) The Covid group seems to be very heterogenous - based on sat O2 values (high SD - 7.5) 

8) The whole draft needs to be revised for clarity and correctness by native speaker. 

9) Please shorten the discussion section - is very redundant and superficial. Please also try to refresh the literature cited since many appropriate papers were published in the Sysmex-based analysis within the last 2 years. 

Comments on the Quality of English Language

Multiple weirdly constructed sentences as well as grammatical errors are present within the whole draft. Native speaker proof reading is required for better understanding of the paper.  

Author Response

I am sending you the attachment.

Reviewer 2 Report

Comments and Suggestions for Authors

The manuscript describes results of analysis of blood by Sysmex XN-1500 automatic system of 3 groups of patients with different diseases and healthy donors. The selected diseases have a significant impact on the healthcare system and the results obtained by the authors could be very interesting for the researchers and medical professionals in the field of infectious diseases, malignancies and lung-specific pathologies.

The main concern about the submitted manuscript is how narrow is the field of Sysmex XN-1500 users and how these studied parameters could be comparable and useful for the regular FACS users. This part should be included in the Discussion.

Specific comments.

- The sentence "The values of size NE-WX, width NE-WY and complexity NE-WZ were the lowest in HC, 27 whereas the highest median proportion of NE-WX, NE-WY and NE-WZ were in LC patients" [27] contradicts the previous statement "Parameters of complexity (NEUT-GI, NE-WX), maturation (IG), size (NE-FSC, NE-WZ) and neutrophil activity (NEUT-RI, NE-WY)" [20]. Please make the corrections accordingly.

- DLCO - abbreviation should be deciphered in Table 1.

- For clear understanding, the authors should be more precise about NEUT-RI parameter in Material and Methods, which "corresponds to RNA/DNA cell content, is plotted on the y-axis and is an indicator of increased RNA activity" in neutrophils. The conclusion that this parameter directly correlates with the metabolic activity of cells is not supported by the data in this manuscript. The changes in NEUTR-RI between different groups of patients were quite small. Such changes could be explained not only by different metabolic status of cells, but could be induced by stress, signaling shift, etc. Authors should introduce analysis of metabolic activity of neutrophils or replace their statements about metabolic activities with RNA content.

Author Response

I am sending you the attachment.

Round 2

Reviewer 1 Report

Comments and Suggestions for Authors

The Authors present a revised version of the paper that was substantially improved and enriched in multiple ways. The Auhtors also responded to most of my majors in an elegant manner as well as provided needed explanations. 

There is still one crucial issue that must be addressed during the second round of the review. The methodology of the data interpretation must be more robustly presented. Namely:

1) Each population/subpopulation on Figure 1 should be clearly labeled. It is not clear to which types of cells given colors match. 

2) Based on Sysmex System manual (Document Number: 1399-LSS, Rev. 3, February 2021) 3 out of 5 analyses in the figure 1 presents "Abnormal WDF Scatter" which is described by the Sysmex team as "needs further validation & confirmation using another lab technique". It also shows that many of the neutrophils might undergo NETosis what might be also reflected by decreased NE count in eg. Covid patients. Also, processing blood samples within 2 hours is not optimal since in volumes <5ml NE start unstimulated NETosis very shortly. 

3) Also, my concern is related to very subjective interpretation of fluorescence results. As Authors stated, there is no specific anti-NE marker used there. As far as I know there is one Sytox-like marker which indicates cell death/membrane permeabilization in each type of cells, including monocytes. It might be a potential bias. 

4) For each significantly changed data representative pictures of the Sysmex results must be given (even in supplementary file). It will decrease substantially the risk of the bias in the study. 

Moreover, did the Authors use the same anticoagulant for each patients' groups? 

Also, Authors say in responses that 2 patients were under mechanical ventilation therapy but table 1 says 3. Please precise. 

In general, I am truly supportive of the publication, albeit after all the concerns regarding methodology are cleared. It is essential to ensure that the presented data is scientifically correct, and paper provides well-verified guidelines for all the clinicians using Sysmex system.

Best regards. 

Round 3

Reviewer 1 Report

Comments and Suggestions for Authors

The Authors revised their manuscript for the second time, again making it much more suitable for the publication. I agree with most of the Authors' reply; however, I still think that the used method is not the best across the laboratory techniques for detection of neutrophils abnormalities.

Nevertheless, I am aware that no further experiments could be done for this paper as well as Authors are aware of some limitations. To make this peer-review process smooth and concise, I would like the Authors to do last, minor adjustments to make sure that the Readers are informed about potential limitations. Please include 4-5 sentences in the discussion during the naming the limitations of the paper, including best-in-class recently published papers from the labs around the world stating that:

-  the results might be slightly impacted by the fact that in the cancer progression NETosis is much more dynamic and abundant compared to healthy state, what impacts the DNA binding staining as well as shape changes of neutrophils - DOI: 10.1158/1078-0432.CCR-24-1363

- in viral infection (eg influenza), neutrophils often form neutrophil-platelets aggregates. Their presence leads most likely to changes in width/height Sysmex parameters of neutrophils which is a cofounding factor for Sysmex data gathering: doi: 10.1172/jci.insight.167299

- lastly, the underlying pharmacotherapy amongst these patients might lead to several neutrophil function impairment, also introducing additional bias to the results.

Also, please consider my comments in your future work.

All the best. 
